# Vitamin B Supplementation and Nutritional Intake of Methyl Donors in Patients with Chronic Kidney Disease: A Critical Review of the Impact on Epigenetic Machinery

**DOI:** 10.3390/nu12051234

**Published:** 2020-04-27

**Authors:** Maria Cappuccilli, Camilla Bergamini, Floriana A. Giacomelli, Giuseppe Cianciolo, Gabriele Donati, Diletta Conte, Teresa Natali, Gaetano La Manna, Irene Capelli

**Affiliations:** 1Department of Experimental Diagnostic and Specialty Medicine (DIMES), Nephrology, Dialysis and Renal Transplant Unit, S. Orsola-Malpighi Hospital, University of Bologna, 40138 Bologna, Italy or maria.cappuccilli@unibo.it (M.C.); giuseppe.cianciolo@aosp.bo.it (G.C.); gabriele.donati@aosp.bo.it (G.D.); diletta.conte2@unibo.it (D.C.); teresa.natali@studio.unibo.it (T.N.); irene.capelli@unibo.it (I.C.); 2Scientific Department, Pharmaelle S.r.l., 40125 Bologna, Italy; bergamini.camilla@pharmaelle.com (C.B.); giacomelli.floriana@pharmaelle.com (F.A.G.)

**Keywords:** chronic kidney disease, cobalamin, DNA methylation, epigenetic, folic acid, methyl donors, vitamin B6, vitamin B12 sublingual formulation

## Abstract

Cardiovascular morbidity and mortality are several-fold higher in patients with advanced chronic kidney disease (CKD) and end-stage renal disease (ESRD) than in the general population. Hyperhomocysteinemia has undoubtedly a central role in such a prominent cardiovascular burden. The levels of homocysteine are regulated by methyl donors (folate, methionine, choline, betaine), and cofactors (vitamin B6, vitamin B12,). Uremia-induced hyperhomocysteinemia has as its main targets DNA methyltransferases, and this leads to an altered epigenetic control of genes regulated through methylation. In renal patients, the epigenetic landscape is strictly correlated with the uremic phenotype and dependent on dietary intake of micronutrients, inflammation, gut microbiome, inflammatory status, oxidative stress, and lifestyle habits. All these factors are key contributors in methylome maintenance and in the modulation of gene transcription through DNA hypo- or hypermethylation in CKD. This is an overview of the epigenetic changes related to DNA methylation in patients with advanced CKD and ESRD. We explored the currently available data on the molecular dysregulations resulting from altered gene expression in uremia. Special attention was paid to the efficacy of B-vitamins supplementation and dietary intake of methyl donors on homocysteine lowering and cardiovascular protection.

## 1. Introduction

It is well known that chronic kidney disease (CKD) confers a markedly increased cardiovascular risk, especially when it progresses to end-stage renal disease (ESRD) [1,2]. Among the non-traditional and uremia-related risk factors accounting for such prominent cardiovascular burden, hyperhomocysteinemia (HHcy) is commonly found in CKD patients because of impaired renal metabolism and excretion [3]. In the general population, as well as in patients with kidney disease, elevated homocysteine levels promote atherosclerosis through increased oxidative stress, impaired endothelial function, and induction of thrombosis (Figure 1).

Homocysteine is regulated by the vitamin B group, namely vitamins B6 (pyridoxine), B9 (folic acid), and B12 (cobalamin). However, despite the beneficial effects of supplementation with these vitamins found on HHcy in the general population, less encouraging and sometimes inconsistent data have been reported in ESRD patients [4,5,6,7,8].

In many cases, the literature has shown that dialysis patients are a peculiar population whose response to certain factors is opposite of that of the general population, a condition that has been referred to as “reverse epidemiology” [9]. An example is represented by hypocholesterolemia, identified as a predictor of increased mortality in dialysis patients [10,11]. Similarly, we have previously demonstrated that higher BMI protects ESRD patients against coronary artery calcifications [12]. In line with this evidence, very low homocysteine levels seem to be associated with worse clinical outcomes, longer hospitalization, and higher all-cause and cardiovascular deaths in ESRD patients [9,13,14]. The combined effect of protein-energy malnutrition and wasting inflammation might partly explain this apparent paradox of the inverse relationship between homocysteine levels and mortality in ESRD patients [14]. However, it remains an open question whether HHcy is a consequence rather than a causal factor of cardiovascular complications in uremic patients, or even just the tip of the iceberg of a complex underlying pathway involving other key molecules [3].

Different experimental and clinical studies highlighted how elevated circulating levels of homocysteine result in intracellular increase in S-adenosylhomocysteine, a competitive inhibitor of methyltransferase, which in turn has been identified as a risk factor of cardiovascular events. Uremia-induced HHcy and the subsequent elevation of intracellular S-adenosylhomocysteine has as main targets DNA methyltransferases (DNMTs), and this leads to altered expression of genes regulated through methylation [15].

Epigenetic changes able to affect CKD and to influence the variability of uremic phenotypes are modulated by several factors, including diet and nutritional status, inflammation, cytokine polymorphisms, gut microbiome, and lifestyle habits [16,17]. In particular, HHcy, inflammatory status and oxidative stress are the key contributors in the modulation of gene transcription through DNA hypo- or hypermethylation in CKD [18,19,20].

This is an overview of the epigenetic changes related to DNA methylation in patients with advanced CKD and ESRD. In particular, we reviewed the current knowledge on the molecular dysregulations resulting from altered gene expression in uremia. A special focus was given to the efficacy of B-vitamins supplementation and dietary intake of methyl donors on homocysteine lowering and cardiovascular protection.

## 2. Methyl-Donor Mediated Epigenetic Effects in CKD

For most genes, DNA hypomethylation commonly results in active transcription, whereas hypermethylation leads to gene silencing since DNA is packaged in an inactive chromatin form.

DNA methylation is catalyzed by DNMTs, an enzyme family responsible for the transfer of a methyl group from S-adenosyl methionine (SAM) to the carbon at position 5 of the cytosine residue of the dinucleotide CpG. On the other hand, gene activation occurs through DNA demethylation catalyzed by dioxygenases of the TET (ten-eleven translocation) protein family composed of TET1, TET2, and TET3 enzymes (Figure 2).

The progression towards renal failure is related to a combination of factors, namely oxidative stress, inflammation, and uremic toxins, that contribute to epigenetic dysregulation. In CKD, a shared interpretation relates genomic hypomethylation to accelerated aging. However, it should also be considered that the interindividual variability of clinical conditions arises from a more complex network, involving promoters and regulatory regions that can be in turn modulated by hyper- or hypomethylation The existing mechanistic link between dietary and epigenetic factors regulates ageing and declining renal function throughout lifetime. In particular, recent evidence has highlighted how epigenetic dysregulation in CKD is dependent on nutrient-sensing pathways. Thus, dietary variations related to socioeconomic conditions or eating habits can differentially affect renal failure progression, related to genomic hypomethylation and inflammatory status. The main factors driving the so-called “inflammaging” may contribute to accelerated senescence and to the establishment of a proinflammatory environment [21,22].

For these reasons, the term “methylome” has been introduced to define the pattern of nucleic acid methylations in the genome or in a specific tissue [23].

The homeostasis of methylome relies on the availability of methyl donors, mainly folic acid, methionine, choline, and betaine. Nevertheless, the impact of their dietary intake on uremic state and ultimately on the epigenetic regulation in CKD is not fully elucidated. Impairments in DNA methylation landscape represent an important molecular feature in several pathologies, including renal diseases [17,24,25,26,27].

In mammalians, DNA methylation is the process by which methyl groups (CH3) are transferred from SAM to the 5-position of cytosine residues to form 5-methylcytosine. This biological pathway is regulated by DNMTs, a highly preserved class of enzymes, with several molecular functions not limited to gene activation and silencing through methylation, but also other post-transcriptional and post-translational modifications [28]. Mammalians’ genomes are generally methylated at CpG dinucleotide sites, thus allowing the binding of proteins and chromatin remodeling factors able to repress transcription. On the other hand, when DNA regions rich in CpG sites, known as “CpG islands”, are hypomethylated, this allows an open chromatin structure, the binding of transcription factors, and the creation of a transcriptionally “permissive” chromatin state [29].

In CKD, the machinery that controls methyl transfer reactions is significantly influenced by the metabolic alterations found in uremic state, since uremic toxins themselves have been proven to be related to methyl metabolism and sulfur amino acid metabolism [30,31].

A prime example of the link between uremia and epigenetic changes occurring in in CKD is MTHFR gene, coding for the enzyme MTHFR, which catalyzes the reduction of 5,10-methylenetetrahydrofolate (methylene THF) to 5-methyl-THF [32,33]. A study by Ghattas et al. on 96 ESRD patients and healthy subjects matched for number, age, sex, and race, reported a higher degree of MTHFR promoter methylation in uremic patients compared to the controls. Moreover, silenced MTHFR gene expression resulted in significantly lower eGFR [34]. Whether this inverse correlation between homocysteine levels and renal function is reversible by dietary and therapeutic interventions is still debated.

In patients who progress towards ESRD and need a renal replacement therapy, the literature has not highlighted substantial differences in terms of methylation profiles with respect to earlier CKD stages.

Hsu et al. investigated the impact of the uremic milieu on DNA methylation and DNA methyltransferase (DNMT) expression, comparing 20 patients under chronic hemodialysis with 20 healthy controls with similar age, gender distribution, and BMI. As expected, dialysis patients showed higher levels of several inflammatory and uremia-related parameters, namely blood urea nitrogen (BUN), serum creatinine, uric acid, calcium, phosphorus, iPTH, cholesterol, CRP, indoxyl sulphate, p-cresol sulphate, and homocysteine. Conversely, the expression of DNMT 1 and 3a did not differ significantly between the groups. Moreover, global DNA methylation was not associated with the levels of uremic toxins and homocysteine, indicating that DNA methylation status may not change significantly in patients undergoing an intensive intervention like hemodialysis [35].

However, the different dialysis techniques seem to affect differently the global DNA methylation status. Ghigolea et al. compared whole blood DNA methylation in 40 on-line hemodiafiltration (HDF) patients vs. 40 high-flux hemodialysis patients vs. 10 control subjects. The results showed a significantly higher methylation degree in hemodialysis patients compared to on-line HDF and controls, suggesting that dialysis technique might affect the degree of inflammation through aberrant DNA hypermethylation [36].

## 3. Effects of Supplementation and Dietary Intake of B-Vitamins and Methyl Donors on Epigenetic Changes in Renal Disease

Nowadays, it is well-assessed that the epigenetic landscape in CKD is strictly correlated with the uremic phenotype and dependent on dietary intake of micronutrients that give an essential contribution to methylome maintenance and epigenetic regulation, as methyl donors (folate, methionine, choline, betaine), or as cofactors (vitamin B12, vitamin B6) [37].

Table 1 summarizes the main dietary sources of folate, vitamin B12, methionine, choline, betaine, and vitamin B6, with the related food content and RDA (Recommended Dietary Allowance).

### 3.1. Folate

The effects of folic acid supplementation to normalize HHcy have been extensively studied in the general population, as well as in CKD patients.

The term “folate” includes different forms of vitamin B9, including tetrahydrofolic acid (the active form), methyltetrahydrofolate (the primary circulating form), methenyltetrahydrofolate, folinic acid, folacin, and pteroylglutamic acid. Since the human body is unable to synthetize folate, it must be supplied by the diet.

Folate is a crucial factor in the production of tetrahydrofolate (THF), a precursor of 5-MTHF required for methionine synthase (MTR) enzyme activity. In the folate cycle, folate transfers one-carbon moieties to different organic compounds, thus controlling SAM levels (Figure 3).

There is a large body of evidence to indicate that folate therapy improves HHcy in the general population, but the data are less clear in CKD patients [8,38,39].

Indeed, the benefits of folate supplementation in subjects with decreased renal function do not seem to lie completely in the lowering of serum homocysteine. An emerging theory is that folate intake improves endothelial dysfunction in CKD, without affecting homocysteine levels [40]. The reason why folate therapy is able to reduce but not normalize HHcy in CKD is not completely understood, but it might be related to the decreased expression of folate receptor 2 and the subsequent impaired sensitivity to homocysteine variations found in uremic patients [39].

A study on 341 hemodialysis patients at our center evaluated the treatment with 5-MTHF (50 mg i.v.) vs. folic acid (5 mg/day oral) and found a better survival rate in patients treated with 5-MTHF, in spite of no significant differences between the groups in terms of HHcy amelioration [7].

In this framework, a key issue is the role of folate supplementation on endothelial dysfunction, another common feature of CKD and an important marker of vascular disease and atherosclerosis. Vascular cells have been proven to be particularly sensitive to HHcy due to their lacking expression of cystathionine β-synthase (CBS), a vitamin B6-dependent enzyme responsible for the hepatic reverse transsulfuration pathway, and betaine-homocysteine S-methyltransferase (BHMT), which controls the alternate betaine-homocysteine remethylation cycle [41,42]. The deficient CBS expression on endothelial cells implies that homocysteine elimination can only occur through the folate and vitamin B12-dependent remethylation pathway under the control of MTHFR and MTS. It has been proposed that renal and vascular injury might share the same mechanism of regarding HHcy and reduced availability of folic acid [42,43,44,45].

In order to clarify the disagreement of the various randomized controlled trials on the effectiveness of folic acid-based homocysteine-lowering therapy on cardiovascular risk protection in renal patients, Qin and colleagues performed two meta-analyses, pooling all the qualified randomized trials from year 1966. In both cases, the results indicated that folic acid therapy is effective in cardiovascular risk reduction by 10%, particularly in studies without grain fortification with folic acid and in those with a lower prevalence of diabetes [46,47].

To summarize the data on folic acid supplementation and the related impact on global DNA methylation status in CKD, the discrepancy in the results of the available studies might be partly explicated by the fact that the paradigm connecting hypermethylation to transcriptional silencing and hypomethylation to gene activation (Figure 1) is not always applicable in CKD. This view, although valid for most genes, should be seen with a critical eye in the complex and multifaceted scenario of renal failure, due to the involvement of uremia-related confounding factors (dyslipidemia, anemia, inflammation, oxidative stress), and to the presence of other genetic elements (promoters, regulatory regions, histone modifications, changes in chromatin conformation, long non-coding RNAs, microRNAs) not directly related to the DNA methylation status. It has been proposed that homocysteine accumulation in blood results in an intracellular rise of S-adenosylhomocysteine (AdoHcy), a strong competitive methyltransferase inhibitor, that has been identified as a predictor of cardiovascular events. Thus, high blood homocysteine levels and high intracellular, commonly found in CKD patients, are associated with abnormal expression of genes regulated through methylation, including imprinted genes and pseudoautosomal genes. However, such alterations can be reversed by homocysteine-lowering therapy [48].

### 3.2. Cobalamin (Vitamin B12)

Vitamin B12, also known as cobalamin, is a nutrient with a key role for human health, being involved as a coenzyme in many chemical reactions essential for human biochemistry. Vitamin B12 deficiency is a common cause of hyperhomocysteinemia and a frequent feature of CKD patients. In one-carbon metabolism, folate serves as the methyl provider and vitamin B12 acts as cofactor for the enzyme methionine synthase, thus folate and vitamin B12 equilibrium is a central regulator of DNA methylation and the epigenetic network, in the general population as well as in nephropathic patients [30,49].

Cobalamin is one of the most complex coenzymes existing in nature: the molecule is constituted by a corrin ring and a dimethylbenizmidazole (DMB) moiety. The focal point of cobalamin structure is the cobalt atom, held at the center of the corrin ring: it can form between four and six bonds, and exists in three oxidation states: Co (III), Co (II), Co (I). Depending on the oxidative state of cobalt, this atom will form a different number of bonds, from six to four; of these, four are always occupied by the nitrogen atoms of the planar corrin ring [50]. The fifth is usually an upper axial ligand (defined as R-group), and in human cobalamin forms, this group usually corresponds to: (i) hydroxo group (hydroxocobalamin, OHCbl); (ii) cyano group (cyanocobalamin, CNCbl); (iii) adenosyl group (adenosylcobalamin, AdoCbl); (iv) methyl group (methylcobalamin, MeCbl).

OHCbl and CNCbl are the most commonly used forms in pharmaceutical formulations for supplementation of vitamin B12. OHCbl is usually found in natural sources and in foods (Figure 3); this form is more unstable compared to CNCbl, but it also binds more tightly to plasma serum proteins (transcobalamin) and consequently persists for a longer time in the bloodstream.

On the other hand, CNCbl, the synthetic form of vitamin B12, is more suitable for pharmaceutical products because of its greater chemical stability [51]. However, both CNCbl and HOCbl were demonstrated to be equally absorbed at pharmacological dosages between 100 and 1000 µg. In target tissues, both OHCbl and CNCbl are metabolized into the two vitamin B12 active forms: AdoCbl in the mitochondria and MeCbl in the cytosol. These molecules are cofactors of several enzymes: they catalyze enzymatic reactions which involve the making and breaking of the Co-C bond of these cofactors. In particular, these two forms are directly required for the function of two enzymes, mitochondrial methylmalonyl-CoA mutase (MUT) and cytosolic methionine synthase (MS) [52,53]. MUT employs AdoCbl to catalyze the conversion of L-methylmalonyl-CoA to succinyl-CoA. This is an indispensable reaction involved in the fatty acid metabolism; additionally, succinil-CoA takes part in the Krebs cycle, a complex of reactions fundamental for cellular energy production. MS, instead, requires the methylated form of cobalamin and catalyzes the remethylation of homocysteine to Met, using 5-methyltetrahydrofolate as the methyl donor. Met produced in this reaction is further converted to SAM; finally, the methyl group of SAM can be donated to form a wide range of vitally important methylated compounds [54]. The tight correlation between MeCbl and folate is accountable for the megaloblastic anemia that occurs in both vitamin deficiencies frequently found in uremic patients, resulting from inhibition of DNA synthesis during red blood cell production [55]. When cobalamin levels become inadequate, DNA synthesis is impaired and cell cycle cannot progress from the G2 growth stage to the mitosis stage; this leads to continuing cell growth without division, leading to macrocytosis.

Given the role of vitamin B12 human metabolic pathways, it is not uncommon to find low cobalamin levels correlated to many different physio-pathological conditions, such as older age, pregnancy, dietary deficiency, bariatric surgery, gastrointestinal diseases, drug treatments, and uremia-related malnutrition. This deficiency is usually caused by malabsorption or reduced intake of foods rich of cobalamin [8,56,57,58].

The impact of cobalamin deficiency on renal function and the benefits of vitamin B12 supplementation have been investigated in several studies. An analysis on a cohort of 2965 subjects from the Framingham Heart Study, replicated in 4445 participants from NHANES 2003–2004, did not highlight a relationship between B12 deficiency, albuminuria, and reduced kidney function [59]. Surprisingly, another study by Soho et al. showed an association between high vitamin B12 levels and all-cause mortality in dialysis patients. The authors explain this unexpected finding in view of the chronic inflammatory status characterizing uremic patients and postulate that this mechanism is an evolutionary adaptation to prevent B12 uptake from infectious organisms in peripheral tissues. Therefore, higher circulating B12 levels may actually reflect functional B12 deficiency in the peripheral tissues, leading to hyperhomocysteinemia and increased cardiovascular risk [60].

However, it is also important to stress that the data on vitamin B12 measurement in serum might be considerably influenced by the form of cobalamin assayed. The standard first-assay to determine vitamin B12 status is the measurement of total serum vitamin B12, a widely available, low cost, automated immune-chemiluminescence-based assay. However, the main limitation of this method lies in the fact that it detects both the inactive forms bound to transcobalamin I and transcobalamin III (referred to as holo-haptocorrin) and the active form of cobalamin in serum (referred to as holo-transcobalamin, which is transcobalamin II-bound). The measurement of total serum vitamin B12 does not allow to discriminate between the active and inactive forms which is an important point, as holo-haptocorrin does not transport cobalamin to cells [61]. Therefore, total serum B12 assay alone is not a reliable biomarker of vitamin B12 status, because around 80% of that is bound to haptocorrin, and not bioavailable for cellular uptake; therefore, patients with strong clinical features of cobalamin deficiency may have serum cobalamin levels that lie within the reference range [62]. In the search of other tests to assess the underlying functional or biochemical deficiency, plasma homocysteine, plasma methylmalonic acid (MMA) and serum holotranscobalamin (holoTC) are the most useful, although not widely used in the current practice, yet [63]. A study by Heil at al. validated on 360 samples the diagnostic accuracy of serum total vitamin B12 and holoTC, confirmed that holoTC can replace total vitamin B12 assay in screening for vitamin B12 deficiency, and determined the clinical decision point for holoTC at 32 pmol/L [64].

The most common signs of vitamin B12 deficiency are hematologic changes and ineffective erythropoiesis [65].

In ESRD, these conditions have been proven to further exacerbate anemia and may increase erythropoietin stimulating agent resistance, the most significant factors found in ESRD population [66]. The only solution in the case of vitamin B12 deficiency is to supplement cobalamin in order to bring serum levels within the physiological range. In general, patients with an irreversible cause should be treated lifelong, while those with a reversible cause should be treated until the deficiency is corrected and symptoms resolved.

Currently, the most used treatment for vitamin B12 deficiency consists of intramuscular cobalamin injections, particularly in dialysis patients due to their compromised intestinal absorption of oral formulations. Approximately 10% of the standard injectable dose of 1000 μg is absorbed, and this allows for rapid replacement in patients with severe deficiency. The therapeutic protocol in the case of important vitamin B12 deficiencies consists of injections of 1000 μg at least several times per week for 1 to 2 weeks, then weekly until clear improvement is shown, followed then by monthly injections. Hematologic response after this attach treatment is fast, with an increase in the reticulocyte count in 1 week and correction of megaloblastic anemia within 6 to 8 weeks.

The use of sublingual delivery is a growing trend in vitamin supplementation. This route of administration avoids the issues of interference with absorption in the lower GI tract, and of the first pass metabolism in the liver, and thus is a feasible alternative to injections [67]. Thanks to this administration route, the active principle is directly picked up in the bloodstream because of the highly vascularized tissue of the sublingual mucosa. Sublingual tablets, then, can produce an immediate systemic effect, allowing drug absorption directly from the oral mucosa, which permits a very fast distribution in the bloodstream and avoids the active principle degradation. Therefore, the sublingual route has great advantages, such as the high bioavailability, fast action and the total absence of pain. At present, available published and unpublished data come from non-renal patients. Strong et al. demonstrated that sublingual treatment of diabetic patients with decreased serum vitamin B12 concentration is as effective as intramuscular injection in correcting a low vitamin B12 status over 6 months [68]. A recent retrospective study carried out by Bensky et al. on more than 4800 patients evaluated the differences between vitamin B12 serum levels after sublingual treatment and intramuscular injections; this study demonstrated that sublingual and intramuscular routes are equally efficient in raising vitamin B12 levels, underlining an even superior efficacy of the sublingual route, which brought to an increase by 193% [69].

The safety and effectiveness of sublingual cobalamin supplementation is under investigation in an ongoing study carried out at five Italian centers (unpublished data). This research pointed out two primary endpoints: first, the evaluation of the response of a malnourished cohort of patients with vitamin B12 deficiency to the administration of sublingual supplementation, with the aim to normalize blood values; the second endpoint considered a cohort of patients who underwent bariatric surgery: due to the well-known difficulties in maintaining appropriate levels of vitamin B12, caused both by a reduced intake and a reduced gut absorption, the aim was to stabilize pre-surgical values. The patients were administered with a long acting sublingual formulation composed of 1000 μg of cobalamin (750 μg cyanocobalamin, 250 μg methylcobalamin) specifically formulated to be administered once weekly. Biochemical parameters, namely hemoglobin, mean corpuscular volume (MCV), hematocrit, homocysteine and total serum vitamin B12, were collected in 195 patients at the time of recruitment (T0) and after 6 months (T1), which is at the first follow up (data from T2, 12 months follow up, are currently being collected).

The first preliminary data arising from this study show an increase of total serum vitamin B12 in 103 patients included up to now, 90.5% of them with a severe baseline vitamin B12 deficiency (<400 pg/mL). After 3 months from the beginning of sublingual supplementations, total serum vitamin B12 was increased by 4.3% in adult patients and 4.4% in the elderly (>65 years). In patients who underwent bariatric surgery, the sublingual supplementation of vitamin B12, once weekly for 6 months, allowed an increase by 17.8% of serum levels, thus halving the cases with severe deficiency.

### 3.3. Vitamin B6

Vitamin B6 is an essential nutrient belonging to the water-soluble B-vitamin group and exists in different forms. The active one, pyridoxal 5′-phosphate, serves as a coenzyme in several enzyme reactions of glucose, lipid, and amino acid metabolism, including the reversible transfer of a one-carbon unit from serine to tetrahydrofolate (THF) to generate glycine and 5,10 MTHF [70,71].

Vitamin B6 can be found in several nutritional sources of animal and vegetal origin, although cooking, storage, and processing might cause significant vitamin B6 losses, depending on the form present in the food [72].

The benefits of vitamin B6 supplementation in CKD have been investigated in the context of combined therapies with other B-vitamins and the available studies report contrasting results. In the previous decade, the HOST and the DIVINe trials failed to prove such positive effects. The HOST study was a large double-blind randomized controlled trial (2001–2006) on patients with HHcy (serum homocysteine levels > 15 mol/L), 1305 of them with advanced CKD (creatinine clearance < 30 mL/min) and 751 with ESRD [44]. The study showed that a daily therapy combining folic acid (40 mg), vitamin B6 (100 mg) and vitamin B12 (2 mg) did not affect survival or incidence of vascular disease, regardless of CKD stage [42]. Successively, the DIVINe trial (Diabetic Intervention with Vitamins to Improve Nephropathy) tested vitamin B therapy (single daily tablet of B-vitamins containing 2.5 mg folic acid, 25 mg vitamin B6, and 1 mg vitamin B12) in 238 participants with diabetic nephropathy from five different Canadian medical centers. Surprisingly, the group of treated patients had lower GFR and higher rate of vascular events compared to the placebo group [73].

Besides, also studies on vitamin D deficiency and insufficiency have revealed a strong association with increased overall mortality and deaths from specific aging-related diseases [74]. Actually, 25(OH)D deficiency has been related to unfavorable clinical outcomes and mortality risk in CKD patients [75]. A study by Obeid et al. investigated the effects of a combined therapy of B-vitamins, vitamin D, and calcium on epigenetic ageing markers [76]. The study was carried out on a limited group of 63 subjects with normal renal function, but it has potential interest in nephrology, in view of the well-known alterations occurring with age in the kidney, namely decline in functional nephron mass, tubulointerstitial changes, glomerular basement membrane thickening, and progressive glomerulosclerosis. The authors analyzed the degree of CpG methylation in three age-correlated genes, aspartoacylase (ASPA), integrin alpha-IIb (ITGA2B), and cAMP-specific 30,50-cyclic phosphodiesterase 4C (PDE4C), comparing two arms of treatment, the first administrated of vitamin D3 + Ca-carbonate and the second with vitamin D3 + Ca-carbonate added with 0.5 mg folic acid, 50 mg vitamin B6 and 0.5 mg vitamin B12. At the end of the one-year follow-up, the three genes showed different behaviors, since methylation of ASPA and PDE4C was higher in the subjects receiving the therapy added with B-vitamins, while methylation of ITGA2B was similar in the two groups. Moreover, in the D + Ca + B treatment arm, lower baseline homocysteine levels were predictive of accelerated aging which was also associated with younger chronological age in both groups [76].

### 3.4. Methionine

Methionine (Met) is an essential sulfur-containing amino acid and a key factor in one-carbon metabolism, as precursor of SAM. Thus, the nutritional intake of Met modulates the SAM/SAH ratio, known as “methylation index” and ultimately the expression of genes regulated by methylation [77]. The main dietary sources of Met are Brazil nuts and animal proteins (meat, fish, eggs, dairy products, and milk). Although Met cannot be completely eliminated from the diet since it is an essential amino acid, there is a large evidence from in vitro, experimental and clinical studies on the benefits of Met restriction in several pathological conditions related to accelerated ageing, including kidney disease [78,79,80,81].

A recent rat model by Pradas and colleagues investigated the renoprotective role of Met restriction, analyzing metabolomic and lipidomic profiling by mass spectrometry in renal cortex specimens from 3 groups of male rats, Adult (8-month-old) vs. Aged (26-month-old) vs. Aged fed with 80% Met restricted diet. The results showed some changes in 59 differential metabolites between Adult and Aged groups in the renal cortex metabolome. Moreover, Met restriction resulted in lipid metabolism reprogramming, mainly affecting glycerophospholipids, docosanoids, and eicosanoids [82].

Another recent experimental study by Wang et al. found a higher survival rate and decreased levels of senescence markers in C57BL/6 mice with a Met restricted diet through a mechanism of upregulation of the transsulfuration pathway, increase of H2S production, and downregulated senescence markers and AMPK (AMP-activated protein kinase) activation [83].

These data are in line with a previous report by Cooke et al. on the effects of Met restriction in young and aged unilateral nephrectomized and 5/6 nephrectomized (5/6Nx) mice. The authors showed that the groups with Met restricted diet had a better trend in terms of renal function parameters (lower urinary albumin, creatinine, albumin-to-creatinine ratio, sulfur amino acids, and electrolytes) and renoprotective biomarkers (clusterin and cystatin C). Moreover, Met restriction induced the activation of genes involved in ion transport (Aqp2, Scnn1a, and Slc6a19), and attenuated kidney injury progression inhibiting inflammatory and fibrosis pathways [84].

Met restriction has been also proven to slow down renal function decline associated to senescence and metabolic dysfunction, improving renal insulin sensitivity, controlling glucose homeostasis, and activating renoprotective genes [85,86].

### 3.5. Choline

Choline is involved in various mechanisms, such as neurotransmitter synthesis (acetylcholine), cell-membrane signaling (phospholipids), lipid transport (lipoproteins), and one carbon metabolism, acting similarly to folate as the methyl provider to support methionine regeneration. Although liver cells are able to synthesize choline, such an endogenous amount is not sufficient for total human requirements. For this reason, choline has been recognized as an essential nutrient, because a supplementary intake by foods is necessary. Modern diet does not seem to provide a sufficient intake of choline (RDA: 550 and 425 mg/day for men and women, respectively) [87,88]. In analogy with other methyl donors, also choline deficiency results in HHcy and epigenetic dysregulation of genes controlled through methylation.

In contrast, CKD patients tend to accumulate choline, because of the impaired choline elimination that physiologically occurs through urine excretion. Hemodialysis is able to remove choline that anyhow returns to baseline levels only after 6 h [89,90].

The extent of choline nutritional intake is a particularly important issue in nephropathic patients, because its blood accumulation triggers a series of metabolic steps controlled by diet, gut microbiome, and host interactions, leading to the release of trimethylamine N-oxide (TMAO), a proatherogenic uremic toxin [91].

The link between gut microbial community, uremic toxins and one-carbon metabolism has been widely investigated [30]. An experimental study by Romano et al. used a murine model with a genetically engineered microbial community lacking for the choline-utilizing enzyme TMA-lyase (CutC). The authors found that mice with choline-consuming bacteria had a decreased bioavailability of this essential methyl-donor, resulting in alterations to host epigenetic regulation. Concerning the impact on renal function, the blood creatine levels were higher in CC+ colonized mice compared with the CC− group, suggesting that impaired choline synthesis compromises creatine formation and leads to renal function decline [92].

### 3.6. Betaine

Betaine is a nonessential nutrient found in various food sources that can also be synthesized from choline and acts as an alternative methyl donor in the remethylation of homocysteine to Met, a reaction catalyzed by the enzyme betaine homocysteine methyltransferase (BHMT). Thus, betaine is involved in the control of the methylation index (SAM/SAH) ratio, and in the epigenetic gene regulation through DNA methylation [24].

In mammalians, betaine is particularly abundant in the kidney, with a main osmoprotective function in medulla cells. It is normally filtered by the glomerulus, and then almost completely reabsorbed in the proximal tubule by cotransport with Na+ or H+ across the luminal plasma membrane [93]. The role of betaine as an osmoprotectant in the kidney occurs through a well-described process triggered by hyperosmotic stress. 

Betaine is present in blood plasma at about 0.1 mM, partly derived from the diet and partly by choline metabolism in the liver. While betaine content in the cortex is due to proximal tubule reabsorption, cells in the kidney medulla uptake betaine primarily from the bloodstream via the sodium and chloride coupled betaine/γ-aminobutyric acid GABA transport system (BGT1). BTG1 is an integral membrane protein sited in the basolateral plasma membrane, predominantly expressed in the thick ascending limb of Henle’s loop and in inner medullary collecting ducts [86]. An in vitro model of cultured MDCK kidney cells showed that BGT1 protein expression can increase more than 10-fold in response to hyperosmotic stress (500 mOsm) and such activation is due to a specific transcription factor, called tonicity-responsive enhancer binding protein (TonEBP) [94,95].

### 3.7. Other Forms of Vitamin B: The Role of Niacin

Nicotinic acid and nicotinamide are collectively referred to as niacin (vitamin B3) and belong to a group of coenzymes that control energy release from carbohydrates and for several redox reactions. They represent the nutritional precursors of nicotinamide adenine dinucleotide (NAD) and nicotinamide adenine dinucleotide phosphate (NADP). NAD and NADP, that serve as cofactors for several cellular redox reactions, thus being essential players in the maintenance of cell metabolism and respiration. There is growing evidence about the relationship between nutritional intake of niacin, NAD(P) availability, and the activity of AD(P)-dependent enzymes that have been proven to be important epigenetic regulators. Niacin intake is essential for the regulation of physiological processes, preservation of genetic stability, and control of epigenetic changes that modulate metabolism and ageing. Thus, niacin plays a crucial role in ageing-related diseases, like cancer, atherosclerosis, metabolic disorders and renal failure [96].

## 4. New Insights in the Network of Methyl Donors, Hyperhomocysteinemia, Epigenetic Landscape, Cardiovascular Risk and CKD Progression

Table 2 details the main clinical trials on the effects of treatment with B-vitamins on homocysteine lowering in patients with various stages of CKD. Since there are several studies on supplementation with folic acid (alone or with vitamins B6 and B12), we preferred to report in the table the results of two successive meta-analyses by Qin et al., collecting the most qualified RCTs studies on the topic published after 1966 [46,47].

The interest on the complex link between methyl donors, HHcy, epigenetic landscape, cardiovascular risk and CKD progression is constantly renewed by the nephrology community.

Over the past few years, research on CKD genetics has moved its focus from single-gene studies towards genome-wide linkage studies and epigenetic landscape.

Among human genome-wide studies, the CRIC (Chronic Renal Insufficiency Cohort) study investigated DNA methylation patterns associated with rapid loss of kidney function by the Infinium HumanMethylation 450 BeadChip, a method to assess the methylation status of 485,577 cytosine positions in the human genome. The study focused on 40 study participants, 20 of them were the ones the highest decline rates in eGFR, and 20 with the lowest. The results showed that genes involved in epithelial to mesenchymal transition and renal fibrosis, namely NPHP4, IQSEC1 and TCF3, were more largely hypermethylated patients with stable kidney function. Other genes of oxidative stress, cell signaling, and inflammatory pathways (NOS3, NFKBIL2, CLU, NFKBIB, TGFB3, TGFBI) were differentially methylated, confirming the regulatory role of epigenetic changes in CKD through activation or suppression of key genes [97].

Given the well-assessed relationship between DNA methylation and CKD, efforts have been made to introduce in the clinical nephrology practice two demethylating agents, 5-azacytidine and decitabine, currently in use for some specific forms of myelodysplastic syndrome and acute myeloid leukemia.

A potential role of 5-azacytidine against renal fibrosis progression has been described by Bechtel et al. in a mouse model of renal fibrosis induced by folic acid. The group investigated the role of hypermethylation of RASAL1 gene that encodes for rasGAP-activating-like protein 1, an inhibitor of the Ras oncoprotein, highly expressed in fibrotic kidneys. The study revealed that hypermethylation resulted in a decreased expression of RASAL1 gene and triggered fibroblast switch to an activated phenotype, persisting also after the mitigation of renal injury. The administration of 5-azacytidine led to a significantly reduced rise in activated fibroblasts, as well as the expression of α-smooth muscle actin and type I collagen. This effect was also confirmed in cultured fibrotic human kidney fibroblasts, as the exposure to 5-azacytidine normalized cell, their proliferative activity as well as the expression of type I collagen and α-smooth muscle actin. An interesting point in this murine model was the differential expression of DNMT gene, as fibrotic mouse kidneys showed an increased DNMT1 expression, but not DNMT3a or DNMT3b expression compared to control kidneys, suggesting a specific role of DNMT1 in RASAL1 methylation and kidney fibrosis progression [98].

Similar findings for hydralazine were recently reported in another murine model of ischemia-reperfusion injury by Tampa et al., aimed at exploring the epigenetic mechanism underlying the development of kidney fibrosis and the AKI-to-CKD evolution. The model confirmed the role of the promoter hypermethylation of RASAL1 gene in the shift from physiological regeneration to tubulointerstitial fibrogenesis. The demethylating agent hydralazine, commonly used as an antihypertensive drug, effectively induced RASAL1 promoter demethylation, thus resulting in attenuated renal fibrosis, improved renal function [99].

Finally, the role of uremic toxins, in particular metabolic products of gut microbiome, deserves special attention, since recent reports have suggested that renal impairment on atherosclerosis is only partially attributable to the elevation of plasma homocysteine. Spence et al. found that the worsening of carotid plaque burden, expressed by total plaque area (TPA), in patients with reduced eGFR, results from a combined effect of HHcy and other uremic toxins, including indoxyl sulfate, p-cresyl sulfate, asymmetric dimethylarginine (ADMA), thiocyanate, trimethylamine and its oxidative product trimethylamine N-oxide (TMAO) [100]. 

Such findings open the way to potential therapeutic and nutritional approaches in an effort to reduce cardiovascular risk in uremic patients. Besides treatment with thiols like mesna to control HHcy [101], there are other interventions like dietary restriction of L-carnitine from red meat and phosphatidylcholine from egg yolks for TMAO reduction [102]. Another useful strategy is represented by overnight daily dialysis, because of its double advantage to normalizes homocysteine levels and to reduce uremic toxins like ADMA, TMAO and other products of gut microbiome [103].

## 5. Conclusions

Epigenetic regulation in patients with CKD and ERSD is strictly related to the interplay of nutritional intake and supplementation of B-vitamins and methyl donors with uremic milieu. Homocysteine is undoubtedly the key factor in methylome status and epigenetic landscape, although the peculiar scenario of uremia is influenced by other confounding factors, like inflammation, gut microbiome, oxidative stress, and lifestyle habits. A better understanding of this complex network may help the clinicians in the challenging riddle of interindividual variations in CKD progression and to identify patient-tailored therapeutic approaches based on their own nutritional deficiencies.

## Figures and Tables

**Figure 1 nutrients-12-01234-f001:**
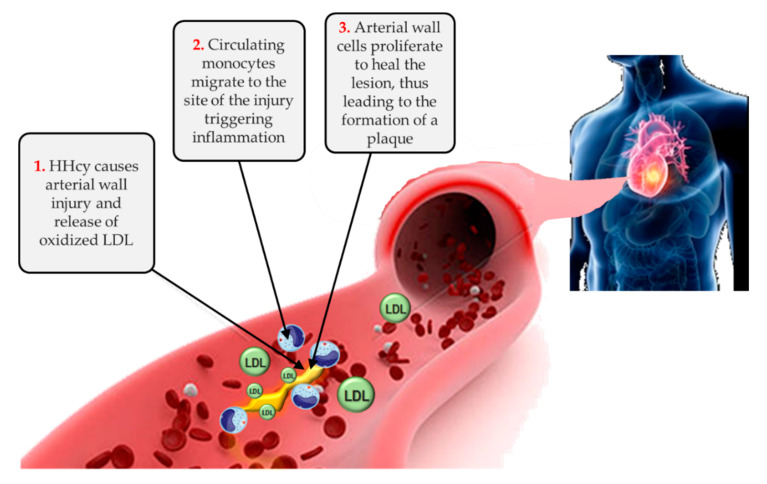
Hyperhomocysteinemia in the atherosclerotic process. HHcy, hyperhomocysteinemia, LDL, low density lipoproteins.

**Figure 2 nutrients-12-01234-f002:**
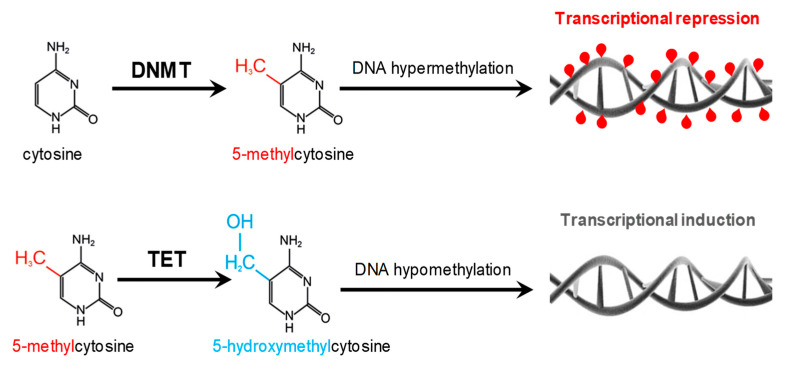
Effects of DNA hypermethylation (**upper**) and hypomethylation (**lower**) on transcriptional activity. DNMT, DNA methyltransferase; TET, Ten-eleven translocation enzymes.

**Figure 3 nutrients-12-01234-f003:**
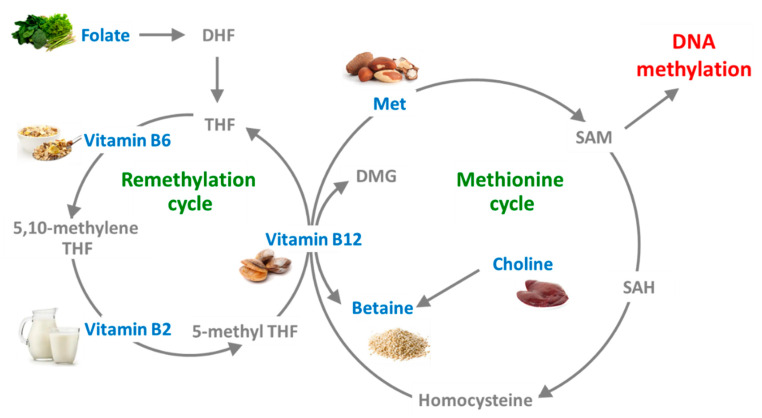
Schematic representation of one-carbon metabolism. DHF, dihydrofolate; DMG, N,N-dimethylglycine betaine; Met, methionine; SAH, S-adenosylhomocysteine; SAM, S-adenosylmethionine; THF, tetrahydrofolate.

**Table 1 nutrients-12-01234-t001:** Main dietary sources of folate, vitamin B12, methionine, choline, betaine, and vitamin B6, related food content, and RDA (Recommended Dietary Allowance).

	Foods	Content	RDA (%)
**Folate**	Asparagus	263 μg/100 g	66
	Cooked spinaches	262 μg/100 g	66
	Cooked lentils	179 μg/100 g	45
	Black eyed peas	179 μg/100 g	45
	Romaine lettuce	114 μg/100 g	29
	Great grains cereals	114 μg/100 g	29
	Cooked broccoli	78 μg/100 g	20
	Sunflower seeds	76 μg/100 g	19
	Fresh orange juice	75 μg/100 g	19
	Cooked beets	69 μg/100 g	17
	Kidney beans	65 μg/100 g	16
**Vitamin B12**	Clams	98.9 μg/100 g	4944
	Liver	85.6 μg/100 g	4280
	Fortified cereals	20.3 μg/100 g	1017
	Mackerel	19 μg/100 g	949
	Beef	7.5 μg/100 g	376
	Crab	6.7 μg/100 g	335
	Swiss cheese	3 μg/100 g	150
	Fortified tofu	1.4 μg/100 g	69
	Eggs (whole)	0.8 μg/100 g	41
	Skimmed milk	0.5 μg/100 g	26
**Vitamin B6**	Cornflakes/branflakes	1.4 mg/100 g	107
	Lean pork meat	0.72 mg/100 g	55.3
	Lean rump meat	0.65 mg/100 g	49.8
	Chicken/turkey breast	0.63 mg/100 g	48.6
	Lamb’s kidney	0.56 mg/100 g	43.1
	Calf’s liver	0.48 mg/100 g	36.9
	Lean minced beef	0.42 mg/100 g	32.3
	Avocado	0.36 mg/100 g	27.7
	Grilled sardines	0.36 mg/100 g	27.7
	Mackerel/plaice	0.27 mg/100 g	20.9
	Pomegranate	10.26 mg/100 g	20.1
**Methionine**	Brazil nuts	1124 mg/100 g	154
	Poultry	931 mg/100 g	128
	Red meat	905 mg/100 g	124
	Tuna	885 mg/100 g	122
	Pork meat	850 mg/100 g	117
	Eggs (whole)	332 mg/100 g	45
	Ricotta cheese	284 mg/100 g	39
	Tofu	211 mg/100 g	29
	Large white beans	146 mg/100 g	20
	Quinoa	96 mg/100 g	13
	Milk	88 mg/100 g	12
**Choline**	Beef liver	350 mg/100 g	63
	Chicken liver	330 mg/100 g	60
	Hard boiled eggs	230 mg/100 g	42
	Smoked salmon	220 mg/100 g	40
	Cooked salmon	91 mg/100 g	17
	Soy protein powder	86 mg/100 g	16
	Roasted chicken	79 mg/100 g	14
	Peanut butter	66 mg/100 g	12
	Almonds	52 mg/100 g	29
	Cruciferous vegetables	40 mg/100 g	7
**Betaine**	Quinoa	630 mg/100 g	97
	Rye	146 mg/100 g	60
	Beets	129 mg/100 g	20
	White bread	102 mg/100 g	16
	Spinaches	89 mg/100 g	14
	Bulgur	83 mg/100 g	13
	Sweet potato	34.6 mg/100 g	5.3
	Veal	33.9 mg/100 g	5.2
	Oat flour	30.7 mg/100 g	4.7
	Tilapia	26.3 mg/100 g	4

**Table 2 nutrients-12-01234-t002:** Effects of treatment with B-vitamins on homocysteine lowering in patients with various stages of CKD. The two meta-analyses by Qin et al. [46,47] collect the most relevant studies published after 1966 about homocysteine-lowering therapy with folic acid in cardiovascular risk prevention in patients with kidney disease.

Study, Year	Design, Duration	Population (n)	Treatment	Homocysteine Decrease	Achievement of Endpoints
Cianciolo, 2008 [8]	Randomized prospective study, 55 months	341 hemodialysis patients	Group A treated with 50 mg i.v. 5-MTHF vs. Group B treated with 5 mg/day oral folic acid (both groups also received i.v. vitamins B6 and B12)	About 50% within the first 6 months of treatment in both groups	Treatment with 5-MTHF reduced inflammation (lower CRP) and increased overall survival rate
Qin, 2011 [46]	Meta-analysis of RCTs from January 1966 to August 2010	3886 patients with ESRD/ACKD from 7 qualified RCTs	Among the 7 selected RCTs, 3 trials used <5 mg of folic acid daily and 4 used ≥5 mg of folic acid daily.	Homocysteine reduction was achieved in the selected RCTs	Folic acid therapy reduced cardiovascular risk in patients with ESRD/ACKD by 15%
Qin, 2013 [47]	Meta-analysis of RCTs from January 1966 to July 2012	8234 patients with kidney disease from 9 qualified RCTs	Folic acid (from 2.5 mg/d to 40 mg/d) alone or with vitamins B6and B12	Homocysteine reduction was achieved in the selected RCTs, but did not significantly correlate with cardiovascular risk	When pooling the 9 RCTs, folic acid therapy reduced cardiovascular risk by 10%
Saifan, 2013 [66]	Short pilot interventional study, 4 months	52 hemodialysis patients	1000 mcg of intramuscular vitamin B12 weekly for the first month and then monthly for 3 consecutive months	B12, homocysteine, and MMA levels were not analyzed as markers of B12 deficiency	Vitamin B12 supplementation resulted in a reduced dose of ESA required to maintain stable hemoglobin levels

Abbreviations: 5-MTHF, 5-Methyltetrahydrofolate; ACKD, advanced chronic kidney disease; CKD, chronic kidney disease; CRP, C-reactive protein; ESA, erythropoietin stimulating agent; ESRD, end stage renal disease; MMA, methylmalonic acid; OL, open label; RCT, randomized controlled trial.

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
