# Peer review of "Vitamin B Supplementation and Nutritional Intake of Methyl Donors in Patients with Chronic Kidney Disease: A Critical Review of the Impact on Epigenetic Machinery"

_nutrients, 2020, doi:10.3390/nu12051234_

Round 1

Reviewer 1 Report

This is an outstanding and very useful review of what is known about vitamin B supplementation and nutritional intake of methyl donors in patients with chronic kidney disease. I have only a few suggestions for enhancing the usefulness of this article.

1. Because haemodialysis is such an intensive intervention, I suggest that ESRD be considered separately from earlier staging of CKD (i.e., as qualitatively distinct rather than merely a quantitative extension of staging) unless the authors present an explicit argument against this. Although haemodialysis is not associated with global differences in methylation profiles (Hsu CY, Sun CY, Lee CC, Wu IW, Hsu HJ, Wu MS. Global DNA methylation not increased in chronic hemodialysis patients: a case–control study. Renal failure. 2012 Nov 1;34(10):1195-9.), the authors should consider the role of haemodialysis on the vascular system.

2. To the extent that the knowledge base permits, the authors should consider other sources of toxicity associated with decreased renal functioning along with the role of methyl donors.

3. Figure 4 Red Meat, not Red Meet

4. p. 6, line 224, be considerably influenced BY the form of cobalamin assayed

5. p. 6, line 231 transport may be a better word choice than vehiculate

6. p. 11, line 394, check reference 42/JamiNson

Author Response

This is an outstanding and very useful review of what is known about vitamin B supplementation and nutritional intake of methyl donors in patients with chronic kidney disease. I have only a few suggestions for enhancing the usefulness of this article.

  1. Because haemodialysis is such an intensive intervention, I suggest that ESRD be considered separately from earlier staging of CKD (i.e., as qualitatively distinct rather than merely a quantitative extension of staging) unless the authors present an explicit argument against this. Although haemodialysis is not associated with global differences in methylation profiles (Hsu CY, Sun CY, Lee CC, Wu IW, Hsu HJ, Wu MS. Global DNA methylation not increased in chronic hemodialysis patients: a case–control study. Renal failure. 2012 Nov 1;34(10):1195-9.), the authors should consider the role of haemodialysis on the vascular system.

This is a reasonable point, although literature does not seem to highlight substantial differences in terms of global DNA methylation status either between early stages CKD and ESRD patients under dialysis, or between HD patients and healthy subjects, as reported in the paper that you suggested (added as ref. 35) and that we discussed in the last sentences of the paragraph “2. Methyl-donor mediated epigenetic effects in CKD”.

In the following sentence, we also described the results of an investigation by Ghigolea and colleagues comparing the impact of different dialysis techniques on global DNA methylation status and revealing a higher methylation degree in hemodialysis patients compared to on-line HDF and normal controls (Ghigolea AB, et al. DNA methylation: hemodialysis versus hemodiafiltration, Ther Apher Dial. 2015, added as ref. 36).

  1. To the extent that the knowledge base permits, the authors should consider other sources of toxicity associated with decreased renal functioning along with the role of methyl donors.

We discussed this point at the end of section 4, that we added in the revised version of the manuscript to underline the novel insights and therapeutic implications.

  1. Figure 4 Red Meat, not Red Meet

The typo has been corrected. However, under request of the reviewer 2, the figures 2-7 have been converted into a table.

  1. p. 6, line 224, be considerably influenced BY the form of cobalamin assayed.

Sorry, it was clearly a careless mistake that we corrected in the revised manuscript.

  1. p. 6, line 231 transport may be a better word choice than vehiculate

Thanks for the suggestion, we addressed this point.

  1. p. 11, line 394, check reference 42/JamiNson

There were some typing mistake in references in square brackets along the text that we have now amended.

Reviewer 2 Report

The manuscript of Cappuccilli et al addresses the role of Vitamin B supplementation and nutritional intake on the epigenetic machinery in patients with chronic kidney disease. In particular, the authors focus on the central role of hyperhomocysteinemia. Hyperhomocysteinemia is thought to be regulated by methyl donor intake as well as cofactors.

This manuscript treats the subject in a superficial manner, providing only the broader context. The authors need to move beyond their current description of the literature and provide novel insight into what is happening. In providing a more detailed examination, looking further into the mechanistic literature, the authors can then subsequently provide genuine insight and perspectives into the subject. As a result, this is a relatively poor manuscript. To improve the manuscript, I would suggest that:

  1. The existing detail should be maintained but condensed by at least 50% to allow space for mechanistic description. – If the mechanistic literature is too sparse, the authors need to bring together the circumstantial evidence and suggest where authors should look to perform mechanistic studies.
  2. The Figures do not currently contribute anything to the literature. There is absolutely no need for the images of the different foods. The data in all the current Figures must be incorporated in one single Table.
  3. Figures should be generated to explain the underlying science – By going into more mechanistic detail, the authors will find material to provide scientific Figures to explain the mechanisms.
  4. The manuscript varies from well written in some sections to very poorly written in others. There are errors of English syntax and grammar throughout the manuscript that need to be corrected by a native speaker. Example sentences that need to be corrected include:
    1. “Abstract: Cardiovascular morbidity and mortality are several-fold increased in patients” should be: morbidity and mortality are increased several-fold
    2. Line 42: HHcy not defined at first use – please check all definitions at first use
    3. “In line with evidences on the reverse biology of dialysis patients” – evidence is always singular, never plural; what is “reverse biology” – this sentence makes no biological sense
    4. “This review is an overview of the epigenetic changes related to DNA methylation in patients with advanced CKD and ESRD that may explicate those molecular dysregulations resulting from altered gene expression in uremia, with a special focus on the efficacy of B-vitamins supplementation and dietary intake of methyl donors on homocysteine lowering and cardiovascular protection.” – This sentence should be broken down into at least 3, if not 4 shorter, simpler sentences.
    5. and continued throughout the manuscript......

Author Response

The manuscript of Cappuccilli et al addresses the role of Vitamin B supplementation and nutritional intake on the epigenetic machinery in patients with chronic kidney disease. In particular, the authors focus on the central role of hyperhomocysteinemia. Hyperhomocysteinemia is thought to be regulated by methyl donor intake as well as cofactors.

This manuscript treats the subject in a superficial manner, providing only the broader context. The authors need to move beyond their current description of the literature and provide novel insight into what is happening. In providing a more detailed examination, looking further into the mechanistic literature, the authors can then subsequently provide genuine insight and perspectives into the subject. As a result, this is a relatively poor manuscript. To improve the manuscript, I would suggest that:

The existing detail should be maintained but condensed by at least 50% to allow space for mechanistic description. – If the mechanistic literature is too sparse, the authors need to bring together the circumstantial evidence and suggest where authors should look to perform mechanistic studies.

We included an additional section before the conclusion describing the recent research on the link between methyl donors, Hhcy, cardiovascular risk and CKD progression, and a more detailed view on the novel insight into CKD genetics, whicn has has moved its focus from single-gene studies towards genome-wide linkage studies and epigenetic landscape.

The Figures do not currently contribute anything to the literature. There is absolutely no need for the images of the different foods. The data in all the current Figures must be incorporated in one single Table.

This point has been addressed in the new version of the manuscript.

Figures should be generated to explain the underlying science – By going into more mechanistic detail, the authors will find material to provide scientific Figures to explain the mechanisms.

The manuscript varies from well written in some sections to very poorly written in others. There are errors of English syntax and grammar throughout the manuscript that need to be corrected by a native speaker. Example sentences that need to be corrected include:

“Abstract: Cardiovascular morbidity and mortality are several-fold increased in patients” should be: morbidity and mortality are increased several-fold

The sentence has been changed according to your suggestion. All the text has been extensively proof-read.

Line 42: HHcy not defined at first use – please check all definitions at first use.

Please note that the acronym had been defined in the 4th line of the introduction, indeed. All the other abbreviations are defined at their first mention.

“In line with evidences on the reverse biology of dialysis patients” – evidence is always singular, never plural; what is “reverse biology” – this sentence makes no biological sense.

“Evidences” has been changed into “evidence”.

Regarding the definition of “reverse biology” in this case referred to dialysis population, we recall previous studies from our and other groups that identified a certain factor that commonly exerts a protective effect  for morbidity and mortality in the general population, as risky in hemodialysis patients (or vice versa).

In the sentence, we specifically mentioned the case of low cholesterol that was found to be predictive of  morbidity and mortality in hemodialysis patients (references 9-11).

As an example, we have defined as “inverse biology” our finding that higher BMI protects ESRD patients against coronary artery calcifications (Cianciolo G et al. Coronary calcifications in end-stage renal disease patients: a new link between osteoprotegerin, diabetes and body mass index? Blood Purif. 2010;29(1):13-22. doi: 10.1159/000245042).

Anyhow, we have modified the text to remove this definition.

“This review is an overview of the epigenetic changes related to DNA methylation in patients with advanced CKD and ESRD that may explicate those molecular dysregulations resulting from altered gene expression in uremia, with a special focus on the efficacy of B-vitamins supplementation and dietary intake of methyl donors on homocysteine lowering and cardiovascular protection.” – This sentence should be broken down into at least 3, if not 4 shorter, simpler sentences.

and continued throughout the manuscript......

According to your suggestion, we have split the very long sentences into 2 or more simpler ones. An extensive proof-reading has also been carried out.

Reviewer 3 Report

In the present paper, Cappuccilli et al review critical literature regarding epigenetic methylation of DNA and it relationship with some vitamin B forms, in the context of chronic kidney disease. The paper is well written and covers most of the data in the area. Here are my suggestions for improvement.

  1. It seems that there are other forms of vitamin B very related with CKD, (like niacin, or even thiamine) could have some effects on the methylation status of DNA. If relevant information is available, it should be included on the text.
  2. I will recommend one or several extra figures to help the reader understand the complex interactions between all the players in the review.
  3. I would also recommend a table including clinical trials on the subject. As clinical trials on vitamin B supplements are unlikely to have as a primary outcome DNA methylation, the risk-benefit of the therapies should be also commented in the text.
  4. On page 15, first paragraph, it is stated that the paradigm of hyper/hypomethylation on DNA transcription is not so clear on CKD due to different confounding factors. This fact is very interesting and should be elaborated further.
  5. Just in the sake of clarity, it should be preferred that all the vitamin-related parts were consecutively explained in the text (now vitamin B6 is at the end).
  6. In pag 10, line 364, there is an extra ‘partly’.
  7. On page 11 there is something wrong with the in-text citations as they display also the first author last name.

Author Response

In the present paper, Cappuccilli et al review critical literature regarding epigenetic methylation of DNA and it relationship with some vitamin B forms, in the context of chronic kidney disease. The paper is well written and covers most of the data in the area. Here are my suggestions for improvement.

It seems that there are other forms of vitamin B very related with CKD, (like niacin, or even thiamine) could have some effects on the methylation status of DNA. If relevant information is available, it should be included on the text.

We added a short section on niacin, although literature lacks specific data on the role of dietary niacin intake in CKD through epigenetic regulation. We tried to highlight the importance of niacin intake in the regulation of physiological processes, preservation of genetic stability, and control of epigenetic changes that modulate metabolism, and ultimately its role in ageing-related diseases, including renal failure. There are no data on thiamine.

I will recommend one or several extra figures to help the reader understand the complex interactions between all the players in the review.

As another reviewer asked to remove the figures detailing the main dietary sources of folate, vitamin B12, methionine, choline, betaine and vitamin B6 with the related food content expressed in RDA (Recommended Dietary Allowance).

We added two extra figures. The current figure 1 illustrates the mechanism by which hyperhomocysteinemia triggers atherosclerosis. The figure 3 illustrates the role of each methyl donor treated in our manuscript in one-carbon metabolism. Moreover, to make the figure 3 clearer and compliant with the journal scope, we also reported the food with the highest content of the various players involved in homocysteine pathway.  

I would also recommend a table including clinical trials on the subject. As clinical trials on vitamin B supplements are unlikely to have as a primary outcome DNA methylation, the risk-benefit of the therapies should be also commented in the text.

We have added a table detailing the most relevant clinical trials and meta-analyses on the role of homocysteine-lowering therapy with B-vitamins in patients with renal disease. The endpoints of each study are also specified.

On page 15, first paragraph, it is stated that the paradigm of hyper/hypomethylation on DNA transcription is not so clear on CKD due to different confounding factors. This fact is very interesting and should be elaborated further.

We have added some sentences to better explain this point, based on the related reference 48.

Just in the sake of clarity, it should be preferred that all the vitamin-related parts were consecutively explained in the text (now vitamin B6 is at the end).

We moved the vitamin B6 section after that of vitamin B12.

In pag 10, line 364, there is an extra ‘partly’.

This was a careless mistake that has been addressed.

On page 11 there is something wrong with the in-text citations as they display also the first author last name.

This point has been addressed.

Round 2

Reviewer 2 Report

This revised manuscript goes a long way to answer my previous concerns.